

# SUMOylation of zebrafish transcription factor Zbtb21 affects its transcription activity

Zhou Fang[1,2,*], Yun Deng[1,2,*], Haihong Wang[1,2] and Jun Zhou[1,2]

[1] Shanghai Institute of Hematology, State Key Laboratory of Medical Genomics, National Research Center for Translational Medicine at Shanghai, Ruijin Hospital, Shanghai Jiao Tong University School of Medicine, Shanghai, China
[2] CNRS-LIA Hematology and Cancer, Sino-French Research Center for Life Sciences and Genomics, Ruijin Hospital, Shanghai Jiao Tong University School of Medicine, Shanghai, China
* These authors contributed equally to this work.

Corresponding authors
Haihong Wang,
wanghaihong51@163.com
Jun Zhou, zj10802@rjh.com.cn

## ABSTRACT

**Background:** Post-translational modification by Small Ubiquitin-like MOdifier (SUMO) is an important mechanism to regulate protein activity, protein stability, and localization of substrates. Zbtb21 is a zinc finger and BTB (Broad-complex, Tram-track and Bric à brac) domain-containing transcription factor. Bioinformatic prediction suggests several putative SUMOylated sites in Zbtb21 protein.
**Methods:** Two evolutionarily conserved lysine residues in Zbtb21 protein were mutated alone or in combination to disrupt the binding with SUMO molecules. Western blot and co-immunoprecipitation analyses were performed to detect the SUMOylation state of wild type and mutant Zbtb21 proteins, respectively. Luciferase reporter assays were conducted to evaluate their transcription activities. Meanwhile, immunofluorescence staining was carried out to show their sub-nuclear localizations. Finally, co-immunoprecipitation was performed to detect the interaction between Zbtb21 and its partners.
**Results:** Phylogenetically conserved lysines 419 and 845 of zebrafish Zbtb21 protein can be conjugated with SUMO molecules. SUMOylation does not affect the subcellular localization and protein stability of Zbtb21, as well as the interaction with Zbtb14 or Zbtb21. Nevertheless, luciferase reporter assays revealed that Zbtb21 is a dual-function transcription factor which exerts activation or repression effect on different promoters, and SUMOylation can modulate the transcriptional activity of Zbtb21 in regulating downstream target genes. Hence, Zbtb21 is identified as a novel substrate of SUMOylation, which would be important for its function.
**Conclusions:** Zebrafish Zbtb21 protein can be SUMOylated on lysines 419 and 845, which is evolutionary conserved. SUMOylation affects the dual role of Zbtb21 on transcription.

# INTRODUCTION

Post-translational modification by Small Ubiquitin-like MOdifier (SUMO) is an important mechanism to regulate many biological processes such as embryogenesis, hematopoiesis,
and tumorigenesis. The process of protein SUMOylation is accomplished by a sequential cascade involving E1 activating enzymes (SAE1/SAE2), E2 conjugating enzyme (UBC9), and E3 ligases, which transfers the SUMO molecules to target lysine residues in specific substrates (*Eifler & Vertegaal, 2015*). Major SUMO paralogs in mammals include SUMO1, SUMO2, and SUMO3. While SUMO1 protein shares only approximately 46% identity with SUMO2/3, the similarity between SUMO2/3 is 95% (*Eifler & Vertegaal, 2015*). The consensus sequence ΨKXE, where Ψ represents a hydrophobic residue, usually contains the SUMOylated lysine of a substrate protein (*Sampson, Wang & Matunis, 2001*). Unlike SUMO1, SUMO2/3 contains a conserved consensus SUMOylation site at the N-terminus, indicating SUMO2/3 are capable to form poly-SUMO chains (*Tatham et al., 2001*).

Accumulating data indicate transcription factors are a major group of SUMO substrates. In most cases described to date, SUMOylation of transcription factors leads to transcriptional repression (*Garcia-Dominguez & Reyes, 2009*; *Valin & Gill, 2007*). However, transcriptional activity of some transcription factors is up-regulated upon SUMOylation (*Kishi et al., 2003*). In addition, SUMOylation also plays various roles in the localization, stability, and protein-protein interaction of transcription factors in a substrate-specific manner, which must be addressed experimentally in each case (*Ouyang, Valin & Gill, 2009*).

There are 49 zinc finger and Broad-complex, Tram-track and Bric à brac (BTB) domain-containing transcription factors in human genome. While the consecutive zinc finger motifs at the C-terminus allow binding with DNA, the BTB motif at the N-terminus mediates the homo/hetero-dimerization/multimerization between different ZBTB proteins (*Maeda, 2016*). Most ZBTB family members function as transcription repressors through recruiting nuclear corepressors such as silencing mediator of retinoid and thyroid hormone receptors (SMRT) and nuclear receptor corepressor (NCoR) (*Maeda, 2016*).

It is worth noting that SUMOylation has important influences on the functions of multiple ZBTB transcription factors. For example, ZBTB16 (also named as PLZF) can be modified with SUMO1 on lysines 242, 387 and 396 in the linker region (*Chao, Chang & Shih, 2007*), which eventually affects its DNA binding capacity or protein stability. ZBTB29 (also named as HIC1) SUMOylation on lysine 314 in the central region favors its interaction with MTA1 (*Dehennaut et al., 2013*). SUMOylation of ZBTB1 on lysines 265 and 328 in the linker region affects its subcellular localization and the binding with SMRT (*Matic et al., 2010*). In addition, ZBTB2, ZBTB9, and ZBTB38 proteins can also be SUMOylated (*Matic et al., 2010*).

In our recent study, zebrafish Zbtb14 was identified as a SUMOylated substrate (*Deng et al., 2022*). SUMOylation of Zbtb14 occurs on a conserved lysine residue within the BTB domain, which renders Zbtb14 capable of repressing *pu.1* promoter during zebrafish macrophage development (*Deng et al., 2022*). A homology search of the BTB domain against the protein database pulled out ZBTB21 as the ZBTB family member with the highest similarity with ZBTB14. Moreover, we noticed that ZBTB21 was described as a direct interactant with ZBTB14 in HEK293T cells and could inhibit multiple genes' promoters (*Wang et al., 2005*). Moreover, bioinformatic prediction (SUMOsp2.0

prediction software) suggested several putative SUMOylated sites in both human ZBTB21 and zebrafish Zbtb21 proteins. Therefore, we speculated that ZBTB21 would be a potential SUMOylated substrate.

Here, we demonstrated that zebrafish Zbtb21 is a novel substrate of SUMOylation, phylogenetically conserved lysines 419 and 845 of Zbtb21 protein can be conjugated with SUMO molecules. SUMOylation does not affect the subcellular localization and stability of Zbtb21, as well as the interaction with Zbtb14 or Zbtb21. However, luciferase reporter assays reveal that Zbtb21 is a dual-function transcription factor that exerts activation or repression effect on different promoters. RNA-seq and ChIP-seq analyses further suggest SUMOylation can modulate the transcriptional activity and DNA binding property of Zbtb21.

## MATERIALS & METHODS

### Plasmid construction

The zebrafish *zbtb21* gene and its serial SUMOylation mutants were cloned into PCS2$^+$ vector (Addgene, #34931, Watertown, MA). For the luciferase reporters, the −1.5 kb promoter of *CDC6* gene and the −1.1 kb promoter of *zbtb14* gene were cloned into the PGL3 basic vector (Promega, Promega, #E1751, Madison, Wisconsin, US). The primers used were listed in Table 1. All primers were designed using the Primer Premier software 5 (PREMIER Biosoft, Palo Alto, CA, US) and synthesized by Sangon (Shanghai, China).

### Cell culture, transient transfection, luciferase reporter assay, and immunofluorescence

HEK293T was obtained from ATCC, and the cell line was tested negative for mycoplasma. HEK293T cells were maintained in DMEM (#11960044, Gibco, Waltham, MA, USA) with 10% fetal bovine serum (#16140071, Gibco, Waltham, MA, USA). Plasmid transfection was conducted utilizing Effectene Transfection Reagent (#301425, Qiagen, Hilden, Germany) following the manufacturer's guidelines.

For the luciferase reporter assay, HEK293T cells were collected 48 h post-transfection and assessed using the Dual Luciferase Reporter Assay Kit (#E1910, Promega, Madison, Wisconsin, USA), in accordance with the manufacturer's instructions.

For microscopic analyses, HEK293T cells were seeded on micro cover glasses in wells of six-well plates 24 h before transfection. 48 h after transfection, cells were immunostained with rabbit anti-HA monoclonal (#3724S, Cell Signaling Technology, Danvers, Massachusetts, USA) at a 1:1,000 dilution, and visualized with secondary antibody, Alexa Fluor 594 goat anti-rabbit IgG (#14708S, Cell Signaling Technology, Danvers, Massachusetts, USA), at a 1:1,000 dilution. Cell nuclei were stained by DAPI (1.0 mg/ml, #564907, BD Biosciences, Dubai, UAE) for 1 min. Fluorescent images were obtained by ECLIPSE E800 microscope (Nikon, Tokyo, Japan) with a SPOT RT Slider digital camera (SPOT Imaging Solutions, Sterling Heights, MI, USA).

**Table 1 Primers used in this work.**

| | Primers for plasmid generation |
|---|---|
| *zf zbtb21* | Forward: 5′-CCGGAATTCACGTCAAATACAAGATGGATG-3′ |
| | Reverse: 5′-CCGCTCGAGTTACGTATGGCTCTGTTCGTG-3′ |
| *zf zbtb21*$^{K419R}$ | Forward: 5′-ACACAGGATTAGAGCAGAGCCCA-3′ |
| | Reverse: 5′-TGGGCTCTGCTCTAATCCTGTGT-3′ |
| *zf zbtb21*$^{K845R}$ | Forward: 5′-GTTGCAGGTTAGAGAGGAACCTC-3′ |
| | Reverse: 5′-GAGGTTCCTCTCTAACCTGCAAC-3′ |
| *hs CDC6-promoter* | Forward: 5′-CGGACGCGTATTCGGATTTGGCGCGAGCG-3′ |
| | Reverse: 5′-CGGCCATGGGACGACAGCACAGCTAGATT-3′ |
| *zf zbtb14-promoter* | Forward: 5′-CGGGGTACCCATCAGTTGTATCTTAGGTACAG-3′ |
| | Reverse: 5′-CCGCTCGAGTGGACTCCTCATGTTTGCTCT-3′ |
| *zf pu.1-promoter* | Forward: 5′-CGGGGTACCACTAGTACACCTAAATTTATG-3′ |
| | Reverse: 5′-CCGCTCGAGATTTGGCAGACCAACAACTGC-3′ |

## Western blot and co-immunoprecipitation assays

HEK293T cells underwent transfection with specified plasmids (#301427, Qiagen, Hilden, Germany). Following 48 h post-transfection, the cells underwent three consecutive 1-min washes with phosphate-buffered saline (PBS) solution. Lysates were generated by treating the cells with RIPA lysis buffer (#P0013J, Beyotime, Jiangsu, China) containing proteinase inhibitor (#5892970001, Roche, Basel, Switzerland), followed by gentle agitation on ice for 30 min. Subsequently, the cells were harvested and centrifuged at $15,000 \times g$ for 30 min at 4 °C. Rabbit anti-HA antibody (#3724S, Cell Signaling Technology, Danvers, Massachusetts, USA) was combined with the supernatant containing protein-G-agarose beads (30 μl) and incubated overnight at 4 °C. Subsequently, the beads were subjected to three washes with RIPA lysis buffer. Finally, proteins attached to the beads were released by the addition of 30 μl of 2 × SDS sample buffer and subsequently analyzed *via* immunoblotting using either anti-SUMO1 (#4930T, Cell Signaling Technology, Danvers, Massachusetts, USA) or anti-FLAG antibody (#F3165, Sigma, Burlington, MA, USA).

## RNA-seq

Forty-eight h after transfection, mRNA was isolated from HEK293T cells by using Trizol reagent (#15596026, Ambion, Austin, Texas, US). Next, mRNA was isolated with Oligo (dT) magnetic beads and fragmented by fragmentation buffer, preparing it for first and second strand cDNA synthesis. After purification of the double-stranded cDNA, the ends were repaired, sequencing adapters were attached, and PCR amplification was performed. The prepared library underwent quality evaluation. Upon confirmation of library quality, sequencing was conducted using an MGI T7 sequencer (China, MGI, Inc). These data are available at https://www.ncbi.nlm.nih.gov/geo/query/acc.cgi?acc=GSE255803.

## ChIP-seq

The EpiTM Chromatin Immunoprecipitation Kit (#R1813, Epibiotek, Seoul, Korea) was utilized for ChIP experiments. Initially, $7 \times 10^6$ fresh cells were fixed with 1% formaldehyde for 10 min, then quenched with 0.125 M glycine for 5 min. The cells were lysed using 1 mL of lysis buffer, and the nuclei were separated by centrifuging the lysate at $2,400 \times g$ for 10 min at 4 °C. The nuclei were then resuspended in digestion buffer and incubated at 37 °C for 10–15 min for enzymatic digestion, fragmenting the chromatin to an average size of 200–500 bp. The chromatin fragments were collected by centrifuging at $18,000 \times g$ for 10 min at 4 °C. The supernatant was then mixed with a ChIP reaction mixture including protein A/G magnetic beads, ChIP IP buffer, antibodies, and protease inhibitors, and incubated with rotation overnight at 4 °C. The following day, the protein A/G magnetic beads underwent washing and removal, while the chromatin was eluted through incubation in reverse cross-linking buffer at 65 °C for 3 h. Subsequently, the ChIP DNA underwent treatment with RNase A and Proteinase K at 37 °C for 30 min, followed by purification using phenol-chloroform extraction. Finally, the ChIP DNA was processed using the QIAseq Ultralow Input Library Kit (#180495, Qiagen, Hilden, Germany) according to the manufacturer's instructions to generate the library. These data are available at https://www.ncbi.nlm.nih.gov/geo/query/acc.cgi?acc=GSE254892.

## Statistical analysis

Data were analyzed by GraphPad Prism 9 using one-way ANOVA for comparisons among multiple groups. Differences were considered significant at $P < 0.05$. Data are expressed as mean ± standard error of the mean (SEM).

# RESULTS

## Zbtb21 is identified as a SUMOylated substrate

Bioinformatic prediction indicated that K419 and K845 are two potential SUMOylated lysine residues of the Zbtb21 protein, located within the linker and zinc finer regions, respectively (SUMOsp2.0 prediction software). To ascertain whether Zbtb21 is indeed a *bona fide* SUMOylated substrate, a series of mutants including Zbtb21$^{K419R}$, Zbtb21$^{K845R}$, and Zbtb21$^{K419/845R}$ (lysine was mutated to arginine to disrupt the attachment with SUMO conjugates) was generated and transfected into HEK293T cells. Western blot analyses revealed four distinct bands corresponding to different forms of Zbtb21 protein.

The lowest band represented unmodified Zbtb21 protein while the other three bands corresponded to SUMOylated forms of Zbtb21 which exhibited increased intensity in the presence of both UBC9 (SUMO conjugating enzyme) and SUMO1 (Fig. 1A, lane 2 and 3). Notably, specific SUMO adducts were absent for Zbtb21$^{K419R}$ and Zbtb21$^{K845R}$ mutant proteins, respectively (Fig. 1A, lane 4–7), whereas no adducts could be observed for Zbtb21$^{K419/845R}$ double mutant (Fig. 1A, lane 8 and 9). These results from western blot analyses were further validated through immuno-coprecipitation (Co-IP) experiments using an anti-SUMO1 antibody (Fig. 1B). While wild type Zbtb21, Zbtb21$^{K419R}$ and Zbtb21$^{K845R}$ single mutants showed successful pull-down of their respective SUMOylated
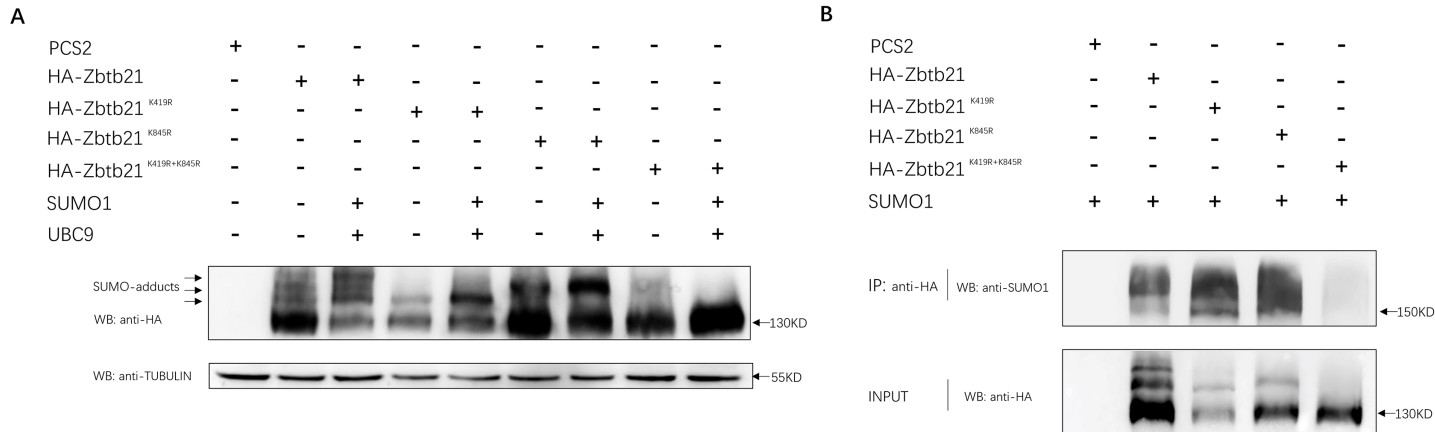

**Figure 1 Zbtb21 is identified as a SUMOylated substrate.** (A) Western blot analyses (anti-HA) of HA-tagged wild type (WT), Zbtb21$^{K419R}$, Zbtb21$^{K419R}$, Zbtb21$^{K845R}$, and Zbtb21$^{K419/845R}$ mutant proteins expressed in HEK293T cells in the absence or presence with the SUMO conjugating enzyme UBC9 and SUMO1. TUBULIN served as internal control. (B) HA-tagged wild type (WT), Zbtb21$^{K419R}$, Zbtb21$^{K419R}$, Zbtb21$^{K845R}$, and Zbtb21$^{K419/845R}$ mutant proteins were immunoprecipitated with an anti-HA antibody from HEK293T cells co-expressing SUMO1, and SUMOylated Zbtb21 protein was detected by western blot with an anti-SUMO1 antibody.

forms upon Co-IP analysis, no such interaction was observed for the Zbtb21$^{K419/845R}$ double mutant.

Taken together, these observations suggest that Zbtb21 is a *bona fide* SUMOylated substrate, with K419 and K845 identified as two lysine residues conjugated by SUMO. Furthermore, the comparable protein levels of wild type Zbtb21 and Zbtb21$^{K419/845R}$ double mutant imply that loss of SUMOylation does not significantly impact the stability of Zbtb21 protein.

## SUMOylation of Zbtb21 affects its transcriptional activity

In most cases, SUMOylation of transcriptional factors is associated with transcription repressing through facilitating interaction with corepressors. *Wang et al. (2005)* demonstrated that ZBTB21 exerted transcriptional repression on the promoter of *CDC6*. Therefore, we conducted luciferase assays using wild-type Zbtb21 and SUMO-defective mutants to investigate their effects on the *CDC6* reporter. Additionally, *zbtb14*, whose upstream regulatory region could be bound by Zbtb21 in our previous study (*Deng et al., 2022*), was also used for luciferase assay. Furthermore, considering that Zbtb14 can inhibit *pu.1* expression (*Deng et al., 2022*), we aimed to explore whether Zbtb21 also possesses repressive function on the *pu.1* promoter.

The results showed that while Zbtb21 exhibited a significant repression effect on the promoters of *CDC6* and *pu.1*, it could activate the *zbtb14* reporter (Figs. 2A–2C), suggesting that Zbtb21 is a context-dependent dual-function transcription factor. Importantly, both the repression and activation activities of Zbtb21 were impaired upon loss of SUMOylation (Figs. 2A–2C).

To gain insights into the global changes in the transcriptome, RNA-seq analyses were carried out in HEK293T cells transfected with equal amounts of wild type Zbtb21 and Zbtb21$^{K419/845R}$ double mutant, respectively (Fig. 2D). Overall, we observed that HEK293T

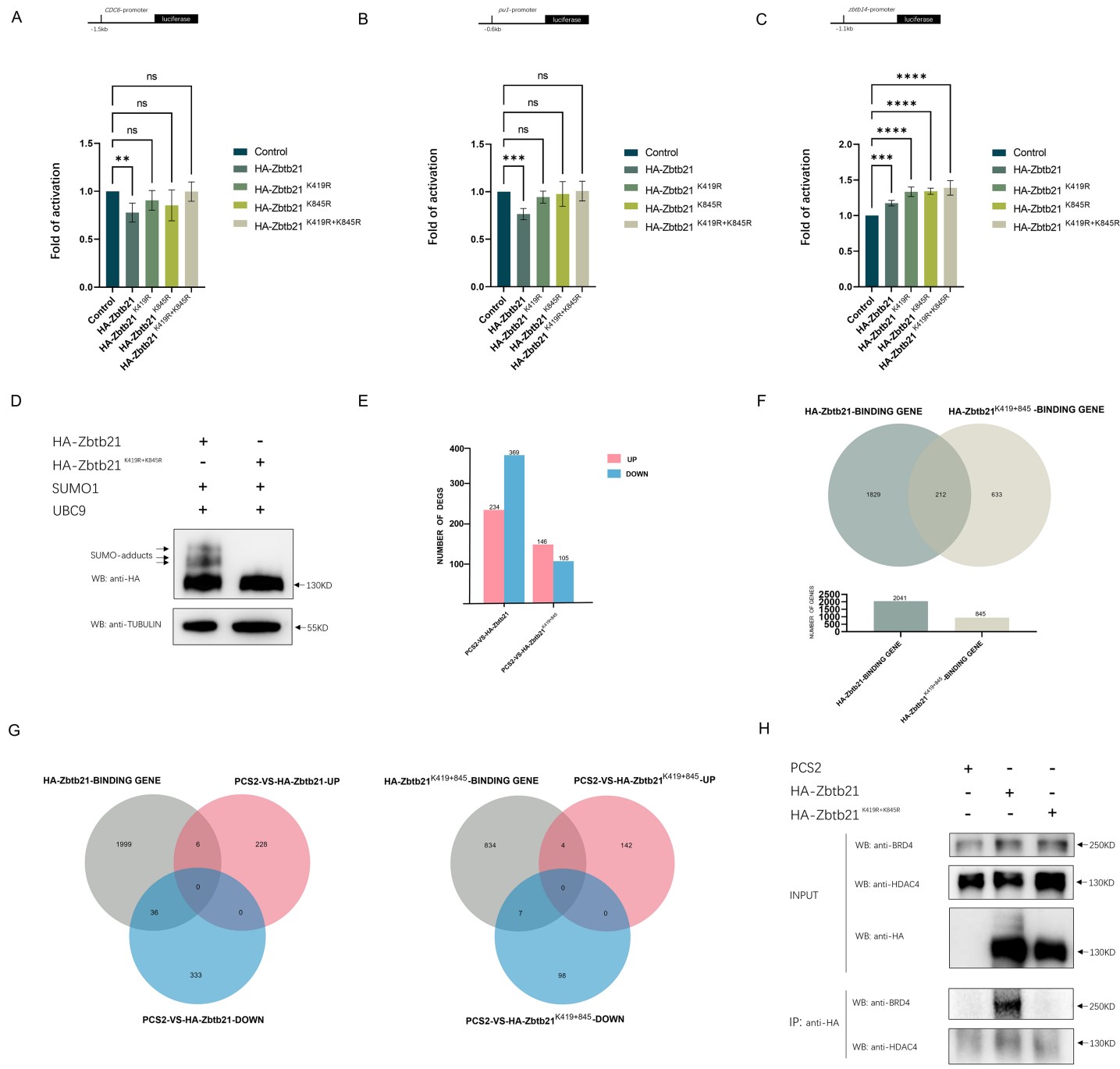

**Figure 2  SUMOylation of Zbtb21 affects its transcriptional activity.** (A–C) Luciferase reporter assays in HEK293T cells expressing HA-tagged wild type (WT), Zbtb21$^{K419R}$, Zbtb21$^{K845R}$, and Zbtb21$^{K419/845R}$ mutant plasmids on *CDC6*, *zbtb14*, and *pu.1* reporters. Bars showed the relative luciferase activity. Luciferase activity was measured after 48 h and normalized to Renilla luciferase (one-way ANOVA, $N = 5$–6. Error bars represent mean ± SEM. ns: not significant, $^{**}P < 0.01$, $^{***}P < 0.001$, $^{****}P < 0.0001$). Specific *P*-values are provided in Table 2. (D) Western blot analysis showed equal amount of wild type Zbtb21 and Zbtb21$^{K419/845R}$ mutant was expressed in HEK293T cells. TUBULIN serves as internal control. (E) Comparison of the differentially expressed genes (DEGs) in HEK293T cells transfected with control, wild type Zbtb21 and Zbtb21$^{K419/845R}$ mutant plasmids, respectively. (F) ChIP analysis showed that while the regulatory regions of 2041 genes were bound by wild type Zbtb21, those of 845 genes were bound by Zbtb21$^{K419/845R}$ mutant. (G) Numbers of upregulated and downregulated genes whose regulatory regions were bound by either wild type Zbtb21 or Zbtb21$^{K419/845R}$ mutant. (H) Co-IP experiments conducted in HEK293T cells expressing HA-tagged Zbtb21 and Zbtb21$^{K419/845R}$. Lysates were immunoprecipitated using anti-HA agarose beads, and analyzed by Western blots with anti-HDAC4 and anti-BRD4 antibodies.

**Table 2 Specific *P*-values for Figs. 2A–2C.**

*zbtb14*-promoter

| PCS2-HA-Zbtb21 | PCS2-HA-Zbtb21$^{K419R}$ | PCS2-HA-Zbtb21$^{K845R}$ | PCS2-HA-Zbtb21$^{K419R + 845R}$ |
|---|---|---|---|
| 0.0007 | <0.0001 | <0.0001 | <0.0001 |

*CDC6*-promoter

| PCS2-HA-Zbtb21 | PCS2-HA-Zbtb21$^{K419R}$ | PCS2-HA-Zbtb21$^{K845R}$ | PCS2-HA-Zbtb21$^{K419R + 845R}$ |
|---|---|---|---|
| 0.0047 | 0.366 | 0.0789 | >0.9999 |

*pu.1*-promoter

| PCS2-HA-Zbtb21 | PCS2-HA-Zbtb21$^{K419R}$ | PCS2-HA-Zbtb21$^{K845R}$ | PCS2-HA-Zbtb21$^{K419R + 845R}$ |
|---|---|---|---|
| 0.0009 | 0.6684 | 0.9742 | 0.9998 |

cells expressing wild type Zbtb21 displayed a total of 234 upregulated genes and 369 downregulated ones (Fig. 2E). However, cells expressing Zbtb21$^{K419/845R}$ mutant showed only 146 upregulated genes and 105 downregulated ones (Fig. 2E). These findings suggest that compared to the SUMOylation-defective mutant, wild type Zbtb21 led to more downregulation of genes, implying that SUMOylation of Zbtb21 favors its transcriptional repression capacity.

Meanwhile, ChIP-seq analyses were conducted to define the genome-wide binding profiles of wild type Zbtb21 and Zbtb21$^{K419/845R}$ mutant. The results showed that while regulatory regions of 2,041 genes were bound by wild type Zbtb21, only 845 genes were found for Zbtb21$^{K419/845R}$ mutant (Fig. 2F). These observations indicate that SUMOylation of Zbtb21 can enhance its DNA binding capacity. Additionally, the regulatory regions of certain sets of genes were bound either by wild type Zbtb21 or Zbtb21$^{K419/845R}$ mutant (1,829 genes *vs.* 633 genes) (Fig. 2F), suggesting that the SUMOylation status can alter the DNA binding property of Zbtb21.

Based on the results from ChIP-seq and RNA-seq analyses, we proceeded to analyze the expression levels of genes whose regulatory regions were bound by wild type Zbtb21 and Zbtb21$^{K419/845R}$, respectively. The results demonstrated that wild type Zbtb21 led to downregulation of 36 genes and upregulation of six genes (Fig. 2G). In contrast, Zbtb21$^{K419/845R}$ mutant showed only seven downregulated genes and four upregulated genes. These findings further support the notion that SUMOylation of Zbtb21 tends to turn it to be a transcriptional repressor (Fig. 2G).

In most cases described to date, SUMOylation of transcriptional factors often leads to transcription inhibition through promoting interaction with corepressors. Notably, histone deacetylase corepressors have been identified as effectors of SUMOylation, playing important roles in transcription regulation (*Yang & Sharrocks, 2004*). Protein interaction database (BioGRID) showed that ZBTB21 can interplay with several corepressors including HDAC1/4/5/7. Therefore, we performed Co-IP analyses in HEK293T cells. Compared to SUMO-defective mutant, more endogenous HDAC4 molecules could be co-immunoprecipitated with wild type Zbtb21 (Fig. 2H), which would be the mechanism by which Zbtb21 inhibits transcription. Additionally, Co-IP analyses also demonstrated

selective binding between wild type Zbtb21 and BRD4 coactivator rather than the SUMO-defective mutant (Fig. 2H).

Collectively, these results indicate that SUMOylation is essential for regulating the function of Zbtb21.

## SUMOylation of Zbtb21 does not affect its subcellular localization

The consequence of protein SUMOylation includes alterations in protein activity, protein stability, and cellular localization (*Chang & Yeh, 2020*). The nuclear localization signal (NLS) of Zbtb21 is close to K419 SUMOylation site. To demonstrate whether SUMOylation plays a role in the subcellular localization of Zbtb21, immunofluorescence staining was performed to compare the localization of wild type Zbtb21 and SUMO-defective mutants. The results showed that HA-tagged wild type Zbtb21 was localized in the nucleus of HEK293T cells, and similar localization was found for Zbtb21$^{K419R}$, Zbtb21$^{K845R}$, and Zbtb21$^{K419/845R}$ mutants (Fig. 3). These observations suggest SUMOylation of Zbtb21 is irrelevant to its nuclear distribution.

## SUMOylation of Zbtb21 does not affect its interaction with its partners

In addition to DNA binding, C2H2-type zinc fingers can mediate protein-protein interactions. The interaction between ZBTB14 and ZBTB21 is mediated by both BTB and zinc finger domains (*Wang et al., 2005*). Since K845 of ZBTB21 is located within the zinc finger region, we set out to investigate whether K845 SUMOylation could affect its binding with ZBTB14. The results from Co-IP assays indicated that the ZBTB21 SUMO mutants could still interplay with ZBTB14 (Fig. 4A). Besides, SUMOylation did not affect the homodimer formation of ZBTB21, either (Fig. 4B).

## DISCUSSION

In the current study, we identified that zebrafish Zbtb21 is a novel substrate of protein SUMOylation, with lysines 419 and 845 serving as two SUMOylated sites. Like many transcription factors, SUMOylation is closely linked with the transcription activity of Zbtb21.These two lysines are conserved across species, implying that modification with SUMO is widely conserved during evolution. Once the two lysines were mutated in human ZBTB21, the SUMO adducts disappeared (Fig. S1). Additionally, endogenous analysis further indicated that ZBTB21 is SUMOylated in HL60 and HEK293T cell lines (Fig. S2).

There are two isoforms for human ZBTB21: ZBTB21L and ZBTB21S. It is worth noting that the two lysines that can be conjugated with SUMO molecules are retained in both long and short isoforms, indicating SUMOylation is important for the function of each ZBTB21 isoform.

There are three major types of SUMOs in vertebrates–SUMO1 and SUMO2/3. Amongst these paralogs, SUMO1 usually modifies its substrates as a monomer (an 11 kD peptide post-translationally attached to target lysine residues), while SUMO2/3 can form poly-SUMO chains, since they possess an intrinsic consensus SUMOylation motif. SUMO2/3 chains can even be capped with SUMO1, which leads to hybrid chain formation (*Chang & Yeh, 2020*). Three SUMO adducts above the unmodified wild type Zbtb21 were observed.

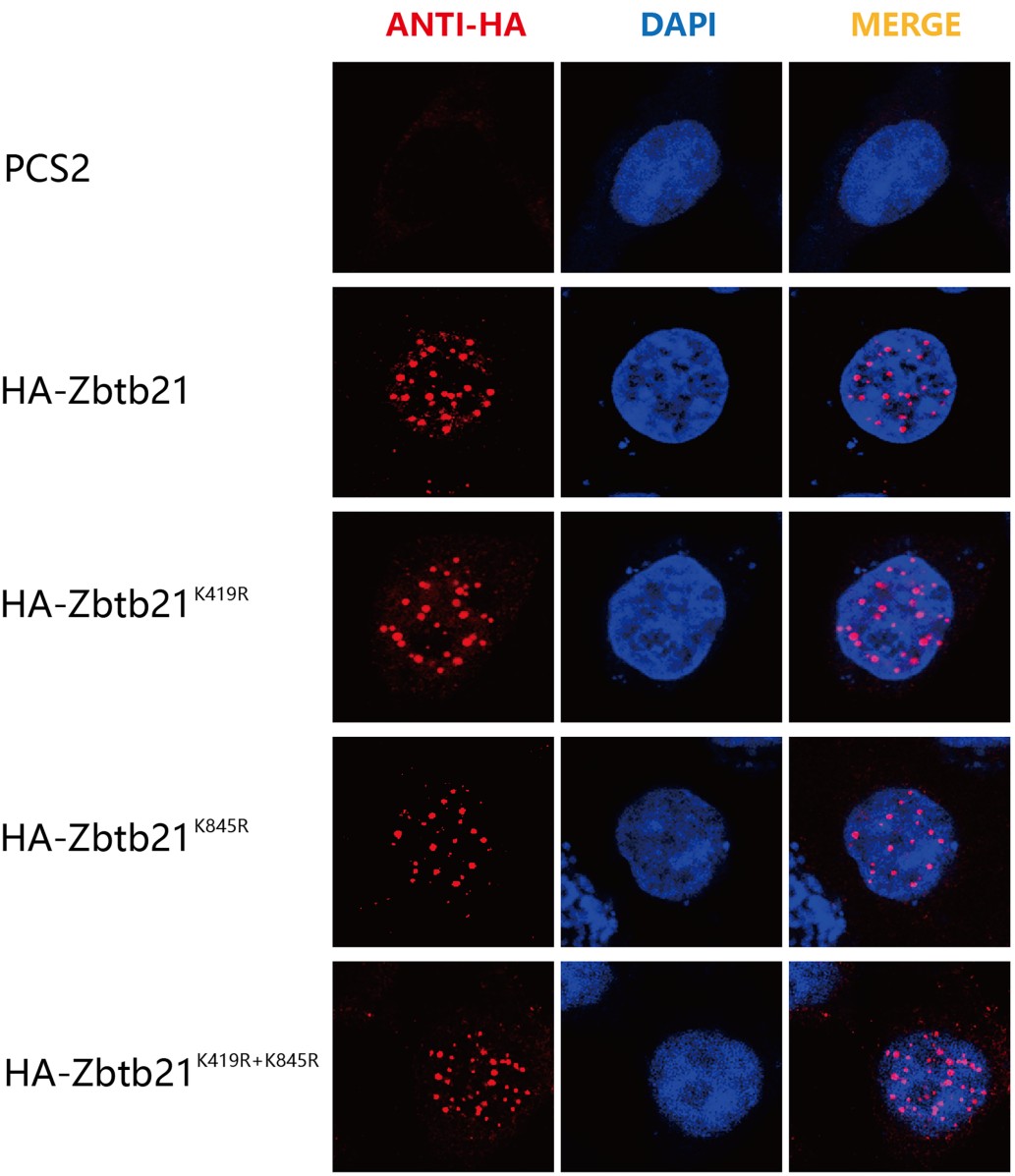

**Figure 3 SUMOylation of Zbtb21 is irrelevant to its subcellular localization.** Immunofluorescence analyses of HEK293T cells expressing HA-tagged WT, Zbtb21$^{K419R}$, Zbtb21$^{K845R}$, and Zbtb21$^{K419/845R}$ mutant proteins. Cells were fixed, and immunofluorescent microscopy was performed using rabbit anti-HA antibody and Alexa Fluor 488 goat anti-rabbit IgG (left). Cell nuclei were stained with DAPI (middle). Digitally merged images are shown on the right side.

While the lowest one represents one SUMO molecule attached covalently with Zbtb21 (which is ~11 kD), the upper two should represent SUMO chains with different lengths on Zbtb21.

While most ZBTB family members function as transcription repressors (*Lee & Maeda, 2012*), ZBTB14 displays an activation or repression effect on different promoters (*Kaplan & Calame, 1997*), and SUMOylation of Zbtb14 turns it to be a potent transcription repressor on the *pu.1* promoter (*Deng et al., 2022*). Although it has been reported that

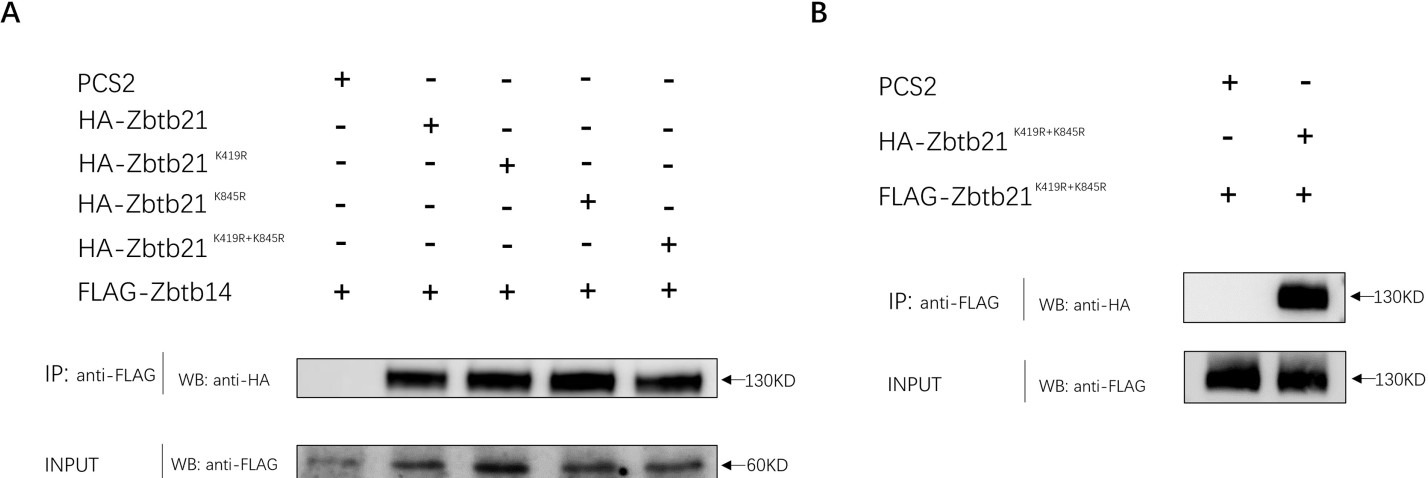

**Figure 4 SUMOylation of Zbtb21 does not affect the hetero-dimerization with Zbtb14 or homo-dimerization with Zbtb21.** (A) Co-IP experiments conducted in HEK293T cell transfected with HA-tagged Zbtb14 and Flag-tagged Zbtb21. Lysates were immunoprecipitated using anti-Flag agarose beads, and analyzed by Western blots with anti-HA antibody. (B) Co-IP experiments conducted in HEK293T cell transfected with HA-tagged Zbtb21 and Flag-tagged Zbtb21 constructs. Flag antibody was applied for immunoprecipitation and HA antibody was used for western blot analysis.                                                                             

ZBTB21 mostly contributes to transcriptional repression (*Wang et al., 2023*), our results show that Zbtb21 also plays a dual role in transcription. While Zbtb21 can inhibit the promoters of *CDC6* and *pu.1*, it activates the *zbtb14* promoter. Interestingly, regardless of activation or repression, SUMOylation of Zbtb21 can enhance its transcriptional activities. Similarly, the zinc finger transcription factor Specificity protein three (Sp3) can activate or repress transcription of target genes in a context-dependent manner, and SUMOylation regulates the duality of Sp3 function (*Valin & Gill, 2007*). Data from RNA-seq showed that less genes were downregulated in the presence of the SUMOylation-defective Zbtb21$^{K419/845R}$ mutant. Likewise, ChIP-seq analyses indicated that the promoters of fewer genes were bound by the Zbtb21$^{K419/845R}$ mutant. These findings suggest that SUMOylation of Zbtb21affects its DNA binding capacity and transcriptional activities. Further studies on the function of Zbtb21 will be helpful to better understand the bi-directional transcriptional regulation in which Zbtb21 is implicated. Recently, *Takebayashi-Suzuki, Uchida & Suzuki (2020)* reported that Zbtb21 cooperates with Zbtb14 to play a crucial function in anterior-posterior patterning during early Xenopus embryogenesis. Hence, SUMOylation of Zbtb21 and Zbtb14 might also play a role in this process, which will be further investigated in the future.

## CONCLUSIONS

ZBTB21 is a novel substrate for SUMOylation. SUMO modification impacts the transcription activities of ZBTB21, particularly its repression function, which is crucial for its role.

## ACKNOWLEDGEMENTS

The authors are grateful to Y Chen (from Shanghai Jiao Tong University School of Medicine, Shanghai, China) for technical support.

### Funding

This work was supported by research funding from the National Natural Science Foundation of China (No. 32171097). The funders had no role in study design, data collection and analysis, decision to publish, or preparation of the manuscript.

### Grant Disclosures

The following grant information was disclosed by the authors:
National Natural Science Foundation of China: 32171097.

### Competing Interests

The authors declare that they have no competing interests.

### Author Contributions

- Zhou Fang conceived and designed the experiments, performed the experiments, analyzed the data, prepared figures and/or tables, and approved the final draft.
- Yun Deng conceived and designed the experiments, performed the experiments, analyzed the data, prepared figures and/or tables, and approved the final draft.
- Haihong Wang conceived and designed the experiments, performed the experiments, authored or reviewed drafts of the article, and approved the final draft.
- Jun Zhou conceived and designed the experiments, authored or reviewed drafts of the article, and approved the final draft.

### Data Availability

The raw data is available at the Supplemental Files.
CHIP-seq and RNA-seq data can be obtained using accession numbers GSE254892 and GSE255803.

### Supplemental Information

Supplemental information for this article can be found online at http://dx.doi.org/10.7717/peerj.17234#supplemental-information.

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
