# Peer review of "SUMOylation of zebrafish transcription factor Zbtb21 affects its transcription activity"

_PeerJ, doi:10.7717/peerj.17234_

## Round 0.1 · original submission · Minor Revisions

Dear Authors,

As per the recommendation of our expert reviewers, I would like to invite you to re-submit a revision of your manuscript. Please do the needful and resubmit ASAP.

Best of luck

·

Basic reporting

The abstract provides a clear and concise overview of the research, including background, methods, and results. It effectively introduces the research topic and the purpose of the study. However, the abstract could be improved by providing a brief summary of the key findings, as it currently lacks a conclusion section that directly states the main outcomes of the research.

Experimental design

The methods section outlines the experimental procedures, such as plasmid construction, cell culture, transfection, and assays. The use of Western blot, co-immunoprecipitation, and luciferase reporter assays demonstrates a well-designed experiment to investigate the SUMOylation of Zbtb21. The inclusion of detailed information about cell lines, antibodies, and reagents used is beneficial for reproducibility. It might be helpful to briefly explain why certain cell lines and reporters were chosen, as well as what the CDC6 and zbtb14 genes are used for in the study.

Validity of the findings

The results suggest that Zbtb21 is indeed SUMOylated, and its SUMOylation affects its transcriptional activity. The study provides evidence supporting these findings through Western blot and co-immunoprecipitation experiments. However, the study lacks an in-depth analysis of the biological significance of these findings.

Additional comments

The manuscript can be accepted after minor changes. All suggestions are included in the PDF file comment box.

Reviewer 2 ·

Basic reporting

Reporting is clear, unambiguous
Sufficiently supported with the studies
Professionally structured the article
However, english and technical language improvement is recommended e.g. use of words like we studied, we reported are not recommended in technical writing, also some minor corrections in tense e.g. indicated in line no. 87 and 184

Experimental design

Original with in the aims and scope of journal
research questions well defined and dealt
methodology also described in detail

Validity of the findings

No comment

Additional comments

Well structured and reported
Experimental Design and result reporting is good
Discussed properly and clearly concluded
Technical writing and language improvement is recommended

Annotated reviews are not available for download in order to protect the identity of reviewers who chose to remain anonymous.

Reviewer 3 ·

Basic reporting

The title is good but it should be more elaborating and exact. The text is written in a good and comprehensive English. Background is very brief. I suggest you to mention the name of the species and the bioinformatic tools being used to carry out the present study. Structure conforms to PeerJ standards. Figures are relevant and labelled. Raw data supplied. All the abbreviations should be expanded. Example: Line 103, 104. It is suggested to mention whether the primers used were defined or designed. If designed, name the software? and if defined, references should be cited in the table. Check the references mentioned in lines 155-156. Add reference in discussion part line 188-189. Check spelling mistake lines 141, 192.

Experimental design

Research findings are novel. Material and methods are well explained and informative to replicate. Original primary research within Scope of the journal. Research question well defined, relevant & meaningful. It is stated how the research fills an identified knowledge gap. Rigorous investigation performed to a high technical & ethical standard.

Validity of the findings

Novelty of the present work has been assessed. The literature is well referenced and also underlying data and figures have been provided to support the findings. However, the conclusion part should be more elaborating. The discussion part needs inclusion of similar and contrary findings for better elucidation of the results obtained. Results of all the experiments taken should be cited in the text as well. Example: Lines 148, 149.

Additional comments

All the comments are covered in 3 sections mentioned above.

Annotated reviews are not available for download in order to protect the identity of reviewers who chose to remain anonymous.

Reviewer 4 ·

Basic reporting

The manuscript investigates the post-translational modification of the transcription factor Zbtb21 by SUMOylation. The authors present evidence that Zbtb21 can be SUMOylated on two conserved lysine residues and that this modification affects the transcriptional activity of Zbtb21 on promoters of different target genes. While the topic is interesting and the experiments appear technically sound, I have some major concerns about the novelty and significance of the findings that should be addressed.

Experimental design

The study design is reasonable for initial characterization of Zbtb21 SUMOylation but lacks the depth and rigor needed to support the transcriptional activity conclusions. Expanding the scope of promoters analyzed and probing the mechanistic basis of transcriptional changes through ChIP and/or co-regulator studies would significantly strengthen the experimental design.

Validity of the findings

Although the identification of Sumolyation sites seems valid, but:

The transcriptional activity effects are only preliminarily demonstrated on 2 promoters.
The small scope limits the validity of claims about SUMO regulation of Zbtb21 function.
No chromatin IP is presented to confirm SUMOylation affects promoter binding as proposed.
There is no investigation of co-regulator recruitment or interactions to support transcriptional mechanisms.
Figure quality is low, with Western blot bands, for example, lacking clear definition.
Suggested Improvements:
To support the idea that SUMO controls transcription, look at promoters on a genome-wide scale and with a larger sample size.
Perform ChIP studies to directly assess effects on promoter binding.
Conduct co-IP or similar experiments to determine co-regulator interactions affected by SUMO.
Enhance the quality of figures for better support of conclusions.

Additional comments

Major Issues:
The introduction lacks clear rationale and background for specifically investigating Zbtb21 SUMOylation. The authors should build a stronger case from the existing literature to justify their focus on Zbtb21.
The writing in the Results section jumps between topics without smooth flow. Improved organization and transitions are needed.
A limitation of only studying overexpressed proteins in HEK293T cells should be acknowledged. Endogenous analysis is preferable.
Minor Issues:
The quality of the figures makes it difficult to interpret the data. Higher-resolution images are needed.
There appear to be formatting inconsistencies in the references. This should be corrected.
There are numerous grammar errors throughout the manuscript text that need to be improved.
In the methods, more details should be provided on critical experimental procedures and reagents used. For example, provide catalog numbers for commercial kits.
Figure legends could be expanded to better explain the assays and conditions tested.
References: Multiple formatting issues throughout the references

Annotated reviews are not available for download in order to protect the identity of reviewers who chose to remain anonymous.

---

## Round 0.2 · Minor Revisions

Dear Authors,

One of our expert reviewers still suggested a few points to be addressed. Therefore, I would like to invite you for minor revisions. Please do the needful and resubmit as soon as possible.

All the best

Reviewer 3 ·

Basic reporting

Corrections incorporated

Experimental design

Corrections incorporated

Validity of the findings

Corrections incorporated

Additional comments

Authors have incorporated all the comments suggested.

Reviewer 4 ·

Basic reporting

Overall Response:
The authors have responded to the major concerns raised in my initial review regarding improving the rationale, experimental support for the transcriptional regulation claims, figure quality, and clarity/organization issues. Adding the pu.1 reporter assay, endogenous co-IP data, enhanced Western blots, extra analyses of human ZBTB21, and a better explanation of the rationale for studying Zbtb21 helped strengthen the work.
However, there are still areas needing improvement that should be addressed before publication:

Experimental design

While expanded by one more gene promoter, the scope of transcriptional regulation analysis is still quite limited. Genome-wide analysis and or more target genes should still be examined to make robust conclusions.
Related to the above, the central finding that SUMO alters Zbtb21 transcriptional regulation is not firmly demonstrated without genomic/expanded studies. Statements claiming SUMO affects activity should be qualified to indicate preliminary evidence.
Sample sizes are still minimal (n=3) for key experiments. Expanding numbers would enhance statistical strength.

Validity of the findings

The Western blots, while improved, could still be better with higher exposures and size standards to interpret band sizes accurately.
Need more information on statistics - what test was used? Actual p values and error bars should be reported on graphs.

Additional comments

Some awkward phrasings still exist. Proofreading to polish language is needed.
Better transition sentences between some paragraphs could help flow and logic.
Overall, the manuscript is much improved but still requires important revisions to demonstrate the key conclusions regarding transcriptional regulation by SUMOylation firmly.

---

## Round 0.3 · accepted · Accept

Dear authors, I am happy to inform you that the manuscript is accepted for the publication. I further request you to be available for few more days to complete various publication tasks.

All the best for your future submissions.

Reviewer 4 ·

Basic reporting

Thank you for submitting this improved version and for your detailed response to the previous critiques. Satisfied

Experimental design

I find the addition of the RNA-seq and ChIP-seq data to be a significant improvement that really enhances the impact of your work. Providing genome-wide analysis of the transcriptional effects and binding patterns of SUMOylated vs non-SUMOylated Zbtb21 was key to supporting your conclusions, and these new results have elevated the rigor and scope of the project. The expanded reporter assays with increased sample numbers and full statistical reporting also strengthen your specific claims.

Validity of the findings

The biochemical data is now of publication quality, with clear blot images and proper documentation of reagents and conditions. I appreciate your efforts to improve figure clarity and expand the legends and methods. These changes make the data much easier to evaluate.

Additional comments

I'm pleased to see that my previous major critiques about experimental scope and data validity have been thoroughly addressed. The manuscript is also significantly easier to read now, with logical organization, smoother transitions between sections, and a more compelling justification for the importance of the work. There are still a couple minor points of awkward phrasing that could be polished prior to publication, but overall the writing is much improved.